# Chemometrics-Assisted Identification of Anti-Inflammatory Compounds from the Green Alga *Klebsormidium flaccidum* var. *zivo*

**DOI:** 10.3390/molecules25051048

**Published:** 2020-02-26

**Authors:** Shi Qiu, Shabana I. Khan, Mei Wang, Jianping Zhao, Siyu Ren, Ikhlas A. Khan, Amy Steffek, William P. Pfund, Xing-Cong Li

**Affiliations:** 1National Center for Natural Product Research, Research Institute of Pharmaceutical Sciences, School of Pharmacy, The University of Mississippi, Oxford, MS 38677, USA; davidhugh@msn.cn (S.Q.); skhan@olemiss.edu (S.I.K.); meiwang@olemiss.edu (M.W.); jianping@olemiss.edu (J.Z.); siyuren07@163.com (S.R.); ikhan@olemiss.edu (I.A.K.); 2Department of Biomolecular Sciences, School of Pharmacy, The University of Mississippi, Oxford, MS 38677, USA; 3ZIVO Biosciences, INC, 2804 Orchard Lake Road, Suite 202, Keego Harbor, MI 48320, USA; asteffek@zivobioscience.com (A.S.); wpfund@zivobioscience.com (W.P.P.)

**Keywords:** *Klebsormidium flaccidum* var. *zivo*, Klebsormidiaceae, anti-inflammatory, iNOS, NF-κB, chemometrics

## Abstract

The green alga *Klebsormidium flaccidum* var. *zivo* is a rich source of proteins, polyphenols, and bioactive small-molecule compounds. An approach involving chromatographic fractionation, anti-inflammatory activity testing, ultrahigh performance liquid chromatography-mass spectrometry profiling, chemometric analysis, and subsequent MS-oriented isolation was employed to rapidly identify its small-molecule anti-inflammatory compounds including hydroxylated fatty acids, chlorophyll-derived pheophorbides, carotenoids, and glycoglycerolipids. Pheophorbide a, which decreased intracellular nitric oxide production by inhibiting inducible nitric oxide synthase, was the most potent compound identified with an IC_50_ value of 0.24 µM in lipopolysaccharides-induced macrophages. It also inhibited nuclear factor kappaB activation with an IC_50_ value of 32.1 µM in phorbol 12-myristate 13-acetate-induced chondrocytes. Compared to conventional bioassay-guided fractionation, this approach is more efficient for rapid identification of multiple chemical classes of bioactive compounds from a complex natural product mixture.

## 1. Introduction

Classical approaches for identification of bioactive compounds from complex natural products involve time-consuming, multi-step bioassay-guided isolation procedures [1,2]. Without appropriate chemical and biological detection methods, often times bioactive compounds may be lost during the isolation process. Over the past decade, however, the application of advanced analytical and chromatographic instrumentation represented by hyphenated techniques such as LC-MS and LC-MS-NMR has greatly improved the efficiency of bioactive natural products discovery [2,3,4]. In addition, data analysis tools utilized in metabolomics and chemometrics have efficiently facilitated the identification of robust chemical markers for differentiation of plant species [5], screening of active compounds [6], and searching for appropriate chemical markers for quality control [7]. 

*Klebsormidium* (Klebsormidiaceae) is a genus of filamentous green algae distributed in terrestrial habitats worldwide [8]. Little is known about the chemistry of the entire genus except for a report on the analysis of fatty acids [9]. *Klebsormidium flaccidum var. zivo* is an optimized, non-GMO (genetically modified organism), proprietary algal strain that produces a unique blend of proteins, polyphenols, and small molecules [10]. The dried algal biomass of *K. flaccidum var. zivo* has been registered as KALGAE^TM^; and a 90-day dietary toxicity study of KALGAE^TM^ in CRL Sprague-Dawley CD IGS rats and a genotoxicity evaluation in Swiss albino mice did not show adverse effects, supporting its safe use as a potential food ingredient to improve human health [10]. However, its small-molecule metabolites associated with biological activities remain unknown. Using an approach involving chromatographic fractionation, anti-inflammatory activity testing, UHPLC-qMS profiling, chemometric analysis, and selective isolation and purification, four chemotypes of compounds including hydroxylated fatty acids, chlorophyll-derived pheophorbides, carotenoids, and glycoglycerolipids, were identified to contribute to the anti-inflammatory activity of this alga. This study highlights the utility of a chemometrics-assisted approach involving orthogonal partial least squares discriminant analysis (OPLS-DA) that can facilitate rapid identification of multiple bioactive compounds from a complex natural product mixture.

## 2. Results and Discussion 

### 2.1. Anti-Inflammatory Activities of Extracts and Column Fractions 

To identify anti-inflammatory compounds from the algal biomass of *K. flaccidum var. zivo*, different extracts using organic solvents and water were prepared to obtain chemical constituents across a wide polarity range. Anti-inflammatory activity of extract was determined in terms of the decrease of inducible nitric oxide (NO) production through inhibition of nitric oxide synthase (iNOS) in lipopolysaccharides (LPS)-induced macrophages (RAW 264.7) and inhibition of nuclear factor kappaB (NF-κB) activation in phorbol 12-myristate 13-acetate (PMA)-induced human chondrosarcoma cells (SW1353) [11]. Preliminary testing results showed that the ethyl acetate extract exhibited highest inhibitory activity in the aforementioned two cell models, indicating that the anti-inflammatory constituents present in this alga are relatively lipophilic small-molecule compounds.

An ethyl acetate extract (IC_50_ 28.0 μg/mL for iNOS) was thus fractionated into 21 fractions, with increasing polarity from Fr. 1 to Fr. 21, by normal-phase column chromatography (Appendix A). Assay for the inhibition of iNOS was used to monitor the anti-inflammatory activity of all 21 fractions. As shown in Figure 1A, the activity is distributed across multiple fractions, indicating the presence of multiple compounds contributing to the activity. The most active column fraction (Fr.10) gave an IC_50_ value of 1.7 μg/mL compared against the IC_50_ value of 0.2 μg/mL of the positive control parthenolide, suggesting potent anti-inflammatory compounds are present in the algal biomass.

### 2.2. Chemical Profiling of Small-Molecule Compounds by UHPLC-qMS-DAD

Reversed-phase ultrahigh performance liquid chromatography (UHPLC) coupled with a single quadrupole mass spectrometer (qMS) and a multi-channel UV detector was utilized to generate chemical profiles of the 21 fractions derived from the ethyl acetate extract with the understanding that the response of different classes of compounds vary considerably under different MS detection methods. Taking advantage of qMS that operates in both positive and negative ionization mode (both ESI and APCI) simultaneously in a single run that facilitates identification of true molecular ion for a specific compound, four classes of compounds, including 21 fatty acid derivatives (**1**–**10**, **12**–**17**, and **20**–**24**), seven pheophorbides (**11**, **26**–**28**, **33**, **39**, and **42**), five carotenoids (**18**, **19**, **25**, **29**, and **30**), and nine glycoglycerolipids (**31**, **32**, **34**–**38**, **40**, and **41**), were detected (Table 1, Figure 1C) [12,13,14,15,16]. A rapid structural identification or prediction of these compounds was conducted as follows.

The presence of fatty acids in *Klebsormidium* spp. [9] prompted a dereplication analysis of such compounds in *K. flaccidum var. zivo*. Typical chromatographic behaviors of major compounds in Fr.1−9 and their precursor ions identified from ESI-MS spectra were clearly associated with fatty acids or their derivatives. Generally, fatty acids are better ionized in negative ESI mode [26]. Hydroxylated fatty acids show de-protonated quasi-molecular ion [M − H]^−^ and de-hydrated protonated quasi-molecular ion [M + H − H_2_O]^+^ in the ESI (−) and ESI (+) MS spectra, respectively. Dihydroxylated fatty acids may produce a fragment ion at *m/z* [M + H − 2H_2_O]^+^ (see Appendix A). Four fatty acids, 7,10,13-hexadecatrienoic acid (**1**), α-linolenic acid (**2**), 8, 11, 14-eicosadienoic acid (**3**), and linoleic acid (**4**), which are present in the least polar Fr. 1 and Fr. 2 were readily determined by the LC-MS data. This was further confirmed by orthogonal GC-MS analysis, which characterized the four compounds and additional non-hydroxylated fatty acids that were not detected by LC-MS (see Appendix A). The remaining 14 hydroxylated fatty acids or their derivatives were characterized at structural type only because LC-MS cannot provide exact structural information of such compounds in terms of positional and configurational double bonds and hydroxyl groups. 

The strong green color of this algal material certainly suggests the presence of chlorophyll-derived compounds. Analysis of the ESI-MS and UV data of the dominant peaks in Fr. 9−11 indicated that pheophorbide a (**26**) and epi-pheophorbide a (**27**) [27] were present in these fractions (Table 1), which was further confirmed by direct comparison of LC-MS-UV data with an authentic sample of pheophorbide a. Partial conversion of **26** into **27** was observed to occur in solution (e.g., methanol) stored at room temperature for a couple of days due to inherent structural tautomerism. Compounds **26** and **27** (*m/z* 593 [M + H]^+^) are characteristic for a strong UV absorption around 410 nm [14]. Additional five analogues with similar UV characteristics were identified in Fr. 6, Fr. 10, Fr. 13, Fr. 17/18, and Fr. 19/20 (Table 1, Figure 1C). Based on their mass data and chromatographic behaviors associated with polarity and retention time, the five compounds were predicted to be pheophorbide a methyl ester (**11**) [14], pheophorbide b (**28**) [14], hydroxyl-pheophorbide a (**33**) [14], isomer of hydroxyl-pheophorbide a (**39**) [14], and hydro-pheophorbide-lactone a (**42**) [25].

Analysis of the compounds with retention time around 23 min (Figure 1C) and characteristic UV absorptions at 430−480 nm in Fr. 6/7 (Table 1) suggested a chemical skeleton for carotenoids as such compounds were previously identified in several algal materials [28]. Lutein (**19**) was unequivocally determined to be present in Fr. 6/7 by direct comparison of LC-MS-UV data with an authentic sample. Zeaxanthin (**18**), a structural isomer of lutein, was also readily identified in Fr. 6/7 by comparison of their UV absorptions in which zeaxanthin has relatively longer UV wavelengths (Table 1, Appendix A) due to an extended conjugated system [15]. The presence of three additional carotenoids, capsanthin (**25**) in Fr. 8/9, and neoxanthin (**29**) and violaxanthin (**30**) in Fr. 11/12, whose structures are closely related to lutein and zeaxanthin, were indicated by their characteristic UV absorptions and molecular weight information (Table 1) [15,18].

Glycoglycerolipids are major components of chloroplast lipids in algae [29]. Our LC-MS data showed the presence of two monogalactosyldiacyglycerols (MGDGs) (**31 [20]** and **32 [21]**) in Fr. 12/13, five monogalactosylmonoacyglycerols (MGMGs) (**34**–**38**) [22,23,24] in Fr. 17, and two digalactosyldiacylglycerols (DGDGs) (**40** and **41**) [12] in Fr. 18. All these galactolipids formed sodium adduct ions [M + Na]^+^ and formate adduct ions [M + HCOO]^−^ in the positive and negative ESI-MS spectra, respectively. In the positive ESI-MS spectra, the fragment ion [M + H – 162 (hexose)]^+^ corresponding to the loss of a galactosyl moiety was observed in MGDGs and MGMGs, while the fragment ion [M + H – 162 × 2]^+^ corresponding to the loss of two galactosyl units were commonly present in DGDGs. For MGDGs and DGDGs, fragment ions corresponding to individual acyl groups, e.g., in compound **31**, were observed as well (Appendix A). 

### 2.3. Determination of Anti-Inflammatory Markers by Chemometric Analysis 

Chemometrics was applied to explore potential anti-inflammatory marker compounds in all 21 column fractions, which were classified into two groups, active and inactive, according to IC_50_ values of iNOS inhibition. Twelve fractions (Fr. 5, Fr. 7–14, and Fr. 17–19) that had enriched activities with IC_50_ values less than that of the parent ethyl acetate extract (28.0 μg/mL) were defined as active, while the remaining nine fractions were considered as inactive (Figure 1A). For each fraction, 654 chromatographic signals generated from the positive ESI-MS were used for OPLS-DA modeling, which clearly showed discrimination of respective fractions in the active and inactive groups (Figure 1B). Thirty-eight signals with variable importance in projection (VIP) values greater than 1.5 were shown from the S-plot (Figure 2A), and were defined as potential anti-inflammatory marker signals. The coefficient plot (Appendix A) indicated that these signals correlated with eight compounds shown in Figure 2B,C. It is interesting to note that the eight marker compounds derived from active fractions represent the aforementioned four classes of compounds. According to the literature, iNOS inhibitory activities of violaxanthin (**30**) (Marker A) [30], pheophorbide a (**26**) (Marker C) [19], and MGDG-1 (**31**) (Marker H) [20] have been reported. Fatty acids, especially hydroxylated fatty acids, have also demonstrated anti-inflammatory activity via multiple molecular mechanisms including iNOS inhibition [31,32], supporting identification of compounds **21** (Marker B) and **16** (Marker F) as anti-inflammatory markers. The remaining three marker compounds **27** (Marker D), **39** (Marker E), and **33** (Marker G) are analogues of pheophorbide a, thereby implicating potential anti-inflammatory activity as well. 

The anti-inflammatory data for all active fractions in terms of IC_50_ values for inhibition of iNOS (as shown in Figure 1A) appear to support that pheophorbide a, the major compound in Fr. 9 and Fr. 10 (Figure 1C, Appendix A), along with pheophorbide a analogues and another three classes of compounds (fatty acids, carotenoids, and glycoglycerolipids) are responsible for much of the anti-inflammatory activity of the extract. Thus, chemometrics is an excellent tool to rapidly predict multiple anti-inflammatory compounds from a complex mixture. 

### 2.4. Isolation and Anti-Inflammatory Testing of Purified Compounds

To validate the prediction of chemometric analysis, Fr. 8 containing fatty acid **21 [16]**, Fr. 10 containing pheophobide a (**26**), and Fr. 12 containing glycoglycerolipid **31** were selected for follow-up isolation of representative anti-inflammatory marker compounds, followed by structural confirmation and anti-inflammatory testing. This selection criterion was based on potent activity, significant quantity, and unique chemical profiles of the active fractions. As a result, marker compounds **21**, **26**, and **31** representing hydroxylated fatty acids, chrolorophyll-derived pheophobides, and glycoglycerolipids, respectively, were isolated, and their structures were confirmed by NMR spectroscopic analysis. The carotenoid violaxanthin (**29**) (Marker A) is a minor compound in Fr. 11 based on the LC-MS-UV profile and its anti-inflammatory activity has previously been reported [30]. Thus, isolation of this compound from Fr. 11 with a relative low activity (IC_50_, 16.0 μg/mL) was not pursued. 

The evaluation of anti-inflammatory activity of the three purified marker compounds indicated that pheophorbide a (**26**) was the most active in inhibiting iNOS with an IC_50_ value of 0.24 µM, while (10*E*,12*Z*,15*Z*)-9-hydroxyoctadecadienoic acid (**21**) and MGDG-1 (**31**) were less active with IC_50_ values of 22.4 and 17.4 µM, respectively (Table 2). Compounds **26**, **21**, and **31** also demonstrated moderate inhibitory activity against NF-κB with IC_50_ values of 32.1, 122.3, and 47.0 µM, respectively. The NF-κB inhibitory activity of **26** had previously been reported [33]. A commercial sample of lutein was also tested, which gave an IC_50_ value of 26.4 µM against iNOS, but was not active against NF-kB (Table 2). These activity data, in conjunction with chemical profiling and relative contents of potential active compounds, explain well the IC_50_ values of relevant active column fractions (Figure 1, Appendix A). Considering the low potency of other active fractions, isolation of additional active compounds from this algal material that outperform pheophorbide a in terms of iNOS inhibition is unlikely. For example, we conducted isolation of hydro-pheophorbide-lactone a (**42**, Figure 3), a structurally close pheophorbide a analogue, which was present in Fr. 19 with an IC_50_ value of 25.0 µg/mL for structural confirmation because only partial NMR data of this compound has been available in the literature [25]. This compound inhibited iNOS with an IC_50_ value of 20.8 µM which is approximately 87-fold less active than pheophorbide a. Thus, pheophorbide a is the most active among all identified anti-inflammatory compounds. 

In conclusion, a chemometrics-assisted approach has demonstrated its advantage in rapidly identifying anti-inflammatory compounds from *K. flaccidum var. zivo* with minimum isolation efforts. Our findings may facilitate product development using an optimized algal biomass with high contents of bioactive compounds. This approach can be utilized to accelerate the discovery of bioactive natural products preferably with novel chemistry.

## 3. Materials and Methods

### 3.1. General Experimental Procedures

The NMR spectra using standard pulse programs were recorded at room temperature on a Bruker Avance DPX-400 spectrometer (Bruker, Billerica, MA, USA) operating at 400 (^1^H) and 100 (^13^C) MHz. The chemical shift (*δ*, ppm) values were calibrated using the residual NMR solvent and coupling constant (*J*) was reported in Hertz (Hz). Column chromatography was done on normal-phase silica gel (230 × 400 mesh, J. T. Baker, Center Valley, PA, USA) or reversed-phase silica gel (C_18_, 40 μm, J. T. Baker). Silica gel 60 F_254_ TLC plates (Merck, Darmstadt, Germany) and reversed-phase TLC plates (C_18_, Merck, Darmstadt, Germany) were used for analytical TLC. The plates were visualized by spraying 10% H_2_SO_4_ followed by heating. The authentic samples pheophorbide a (>95%) and lutein (>90%) were purchased from Cayman Chemical Company (Ann Arbor, MI, USA) and Acros Organics (Pittsburgh, PA, USA), respectively. Their structure and purity were further confirmed by NMR and HPLC in our laboratory. The 37 standards of fatty acid methyl esters (Supelco® 37 components FAME mix) used for GC-MS analysis were purchased from Sigma-Aldrich (St. Louis, MO, USA). 

### 3.2. Plant Material

The algal biomass KALGAE^TM^ is derived from *K. flaccidum* var. *zivo* and its general physical, chemical, and nutritional compositions has been descried in a previous paper [10].

### 3.3. Extraction of Algal Biomass 

In total, 10.0 g of powdered biomass was extracted with 200 mL of methanol under ultrasonication (40 kHz, 300W) for 1 h, and allowed to stand overnight. The mixture was then filtered using a vacuum suction filter to obtain the organic phase. A portion of organic phase (50 mL) was evaporated to dryness to yield a methanol extract (0.41 g). The remaining 150 mL was evaporated to dryness, suspended in 150 mL of water and extracted with an equal volume of ethyl acetate five times. The combined ethyl acetate layers were concentrated to dryness to yield an ethyl acetate extract (0.36 g). Scale-up extractions with same ratios of sample and solvent produced similar yields of extracts. To prepare the water extract that may contain polysaccharides, 10.0 g of powdered biomass was extracted with 200 mL of water under ultrasonication (40 kHz, 300W) for 1 h. The mixture was then filtered using a vacuum suction filter to obtain the water phase, which was freeze dried to yield the desired extract (2.44 g). 

### 3.4. Fractionation of Ethyl Acetate Extract

15.0 g of ethyl acetate extract, prepared from a scale-up extraction, was absorbed on normal-phase silica gel and then loaded on a normal-phase silica gel column (200 g). The column was eluted with chloroform first and then a gradient solvent system consisting of chloroform and methanol in ratios of 100:1, 50:1, 20:1, 10:1, 5:1, 1:1. Based on the TLC analysis, fractions containing similar compounds were combined to afford 21 major fractions (Fr.1−21). The detailed information on weight and solvent systems used to generate fractions is shown in the Appendix A.

### 3.5. Chemical Profiling of Column Fractions by UHPLC-qMS-DAD

An Agilent Series 1290 UHPLC system (Agilent Technologies, Santa Clara, CA, USA) coupled with a DAD detector and an Agilent 6120 quadrupole-MS mass spectrometer were utilized for chemical profiling. Column fractions were prepared at a concentration of 1 mg/mL in methanol. Separation was achieved using an Agilent Eclipse Plus-C_18_ column (100 × 2.1 mm, 1.8 µm) maintained at 25 °C. Binary mobile phase was 0.1% formic acid in water (A) and 0.1% formic acid in acetonitrile (B) at a flow rate of 0.25 mL/min under the following gradient program: 0–6 min, 30–50% (B); 6–10 min, 50–80% (B); 10–23 min, 80–95% (B); 23–25 min, 95–100% (B). The UV wavelength detection was set as 254, 400, 410 and 450 nm with a bandwidth of 4. The spectrum was stored from 190 to 600 nm. MS data was obtained under the ESI (±) mode. Optimal source parameters were set as follows: drying gas (N_2_) flow rate, 10.0 L/min; drying gas temperature, 300 °C; nebulizer, 35 psig; vaporizer temperature, 200 °C; capillary voltage, 4.0 kV; fragmentor voltage, 100 V. Each sample was analyzed over the mass range of *m/z* 100-1500. A QC sample was prepared using pooled column fractions and analyzed once every three tested samples to monitor the system stability. 

### 3.6. Chemometric Analysis 

Raw UHPLC-qMS data of 21 column fractions acquired in the ESI (+) mode were processed by Agilent ChemStation software (E.02.02) to export a data matrix containing retention times and peak intensities into Excel, in which 654 signals of each fraction were included in the retention time window between 5 and 25 min. This dataset was used as an input for SIMCA-P 13.0 to perform OPLS-DA. The corresponding VIP values were calculated. According to the distribution of signals and VIP values, the signals showing a VIP value > 1.5 were considered as potential marker signals. The marker signals were then correlated with specific compounds based on the retention time and LC-qMS identification of active compounds in column fractions.

### 3.7. Isolation of Anti-Inflammatory Marker Compounds from Active Column Fractions

Detailed procedures for the isolation of compounds **21**, **22**, and **24** from Fr. 8, **26** from Fr. 10, **31** from Fr. 12, and **42** from Fr. 19 are shown in Appendix A.

### 3.8. In Vitro Anti-Inflammatory Assays for Inhibition of iNOS and NF-κB

Extracts (20 mg/mL), fractions (20 mg/mL) and compounds (2 mg/mL) were dissolved in DMSO and diluted in serum free media prior to the assay to achieve various test concentrations. The assays for inhibition of iNOS and NF-κB activity were performed in mouse macrophage (RAW264.7) and human chondrosarcoma (SW1353) cell lines, respectively. Both cell lines were obtained from American Type Culture Collection (ATCC), Manassas, VA, USA. The detailed procedures have been described in a previous publication [34]. IC_50_ values were obtained from concentration response curves. Parthenolide was used as a positive control in the both assays.

## Figures and Tables

**Figure 1 molecules-25-01048-f001:**
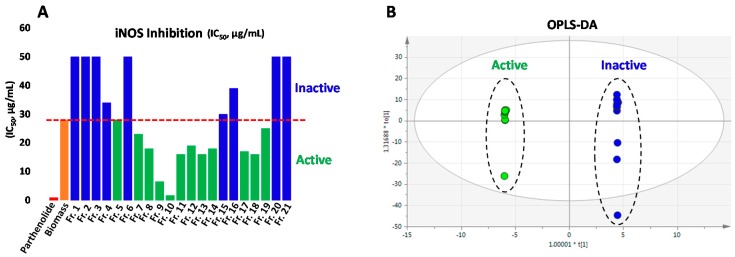
(**A**) Anti-inflammatory activities of 21 column fractions derived from *K. flaccidum var. zivo* against iNOS, in comparison with the parent ethyl acetate extract and the positive control parthenolide; (**B**) Orthogonal partial least squares discriminant analysis (OPLS-DA) discriminating active and inactive fractions; and (**C**) UHPLC-qMS profiling of the ethyl acetate extract and its 21 column fractions with compound numbers labeled beside respective peaks.

**Figure 2 molecules-25-01048-f002:**
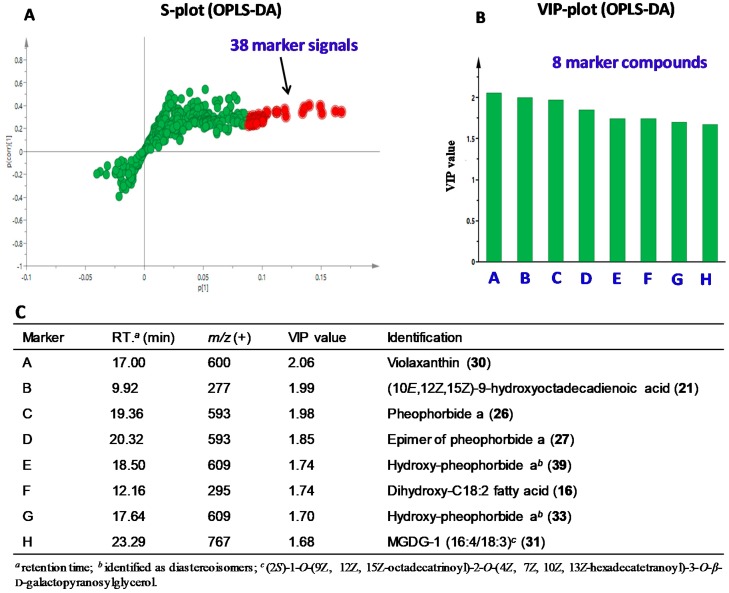
Prediction of anti-inflammatory marker compounds by chemometrics: (**A**) OPLS-DA S-plot of chromatographic signals of 21 column fractions generated from the positive ESI-MS with 38 marker signals (VIP > 1.5) highlighted in red; (**B**) VIP plot showing 8 marker compounds corresponding to 38 marker signals; and (**C**) chromatographic and MS information of the 8 marker compounds.

**Figure 3 molecules-25-01048-f003:**
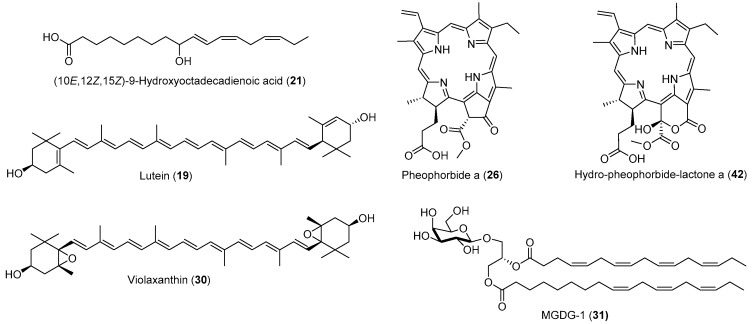
Structures of anti-inflammatory compounds from *K. flaccidum var. zivo*.

**Table 1 molecules-25-01048-t001:** Compounds **1**–**42** detected by UHPLC-qMS-UV in column fractions.

Compound	*m/z* (+) *^a^*	*m/z* (-) *^b^*	RT *^c^* (min)	UV_max_ (nm)	Location *^d^*	Identification/Prediction	Reference
**1**	251	295	13.18	\	Fr. 2	7, 10, 13-Hexadecatrienoic acid *^e^*	[12]
**2**	279	323	15.92	\	Fr. 2,3,4,5	α-Linolenic acid *^e^*	[13]
**3**	305	349	17.60	\	Fr. 2	8, 11, 14-Eicosadienoic acid *^e^*	[12]
**4**	281	325	17.98	\	Fr. 2	-Linoleic acid *^e^*	[12]
**5**	277	293	11.77	\	Fr. 4,5	Hydroxy-C18:3 FA *^f^*	
**6**	279	295	13.16	\	Fr. 4,5	Hydroxy-C18:2 FA *^f^*	
**7**	281	301	14.96	\	Fr. 4	Hydroxy-C18:1 FA *^f^*	
**8**	283	299	19.74	\	Fr. 4	Hydroxy-C18:0 FA *^f^*	
**9**	305	321	9.67	\	Fr. 6	Hydroxy-C20:3 FA *^f^*	
**10**	307	323	10.83	\	Fr. 6	Hydroxy-C20:2 FA *^f^*	
**11**	607	605	22.51	408	Fr. 6	Pheophorbide a methyl ester	[14]
**12**	293	309	10.47	\	Fr. 7	Dihydroxy-C18:3 FA *^f^*	
**13**	293	309	10.71	\	Fr. 7	Dihydroxy-C18:3 FA *^f^*	
**14**	295	311	11.78	\	Fr. 7	Dihydroxy-C18:2 FA *^f^*	
**15**	295	311	11.98	\	Fr. 7	Dihydroxy-C18:2 FA *^f^*	
**16**	295	311	12.16	\	Fr. 7	Dihydroxy-C18:2 FA *^f^*	
**17**	295	311	12.42	\	Fr. 7	Dihydroxy-C18:2 FA *^f^*	
**18**	568	\	22.95	454; 478	Fr. 6,7	Zeaxanthin	[15]
**19**	568	\	23.18	448; 474	Fr. 6,7	Lutein	[15]
**20**	277	293	9.71	\	Fr. 8,9	Hydroxy-C18:3 FA *^f^*	
**21**	277	293	9.92	\	Fr. 8	(10E,12Z,15Z)-9-hydroxyoctadecadienoic acid	[16]
**22**	279	295	11.05	\	Fr. 8,9	(10E,12Z)-9-hydroxyoctadecadienoic acid	[16]
**23**	278	\	13.12	\	Fr. 8	C18:3 FAA *^g^*	
**24**	280	\	14.98	\	Fr. 8	(9Z,12Z)-octadecadienamide	[17]
**25**	584 *^h^*	\	17.59	448; 474	Fr. 8,9	Capsanthin	[18]
**26**	593	591	19.36	410	Fr. 9,10,11	Pheophorbide a	[19]
**27**	593	591	20.32	410	Fr. 9,10,11	15-Epimer of Pheophorbide a	[19]
**28**	607	605	15.86	436	Fr. 10	Pheophorbide b	[14]
**29**	600 *^h^*	\	16.55	438; 466	Fr. 11,12	Neoxanthin	[15]
**30**	600 *^h^*	\	17.00	440; 471	Fr. 11,12	Violaxanthin	[15]
**31**	767	789	23.29	\	Fr. 12,13	MGDG *^i^*(16:4/18:3)	[20]
**32**	769	791	24.85	\	Fr. 12	MGDG (16:3/18:3)	[21]
**33**	609	607	17.64	408	Fr. 13	15-Hydroxy-pheophorbide a *^j^*	[14]
**34**	507	529	6.99	\	Fr. 17	MGMG *^k^* (C16:4)	[22]
**35**	509	531	7.66	\	Fr. 17	MGMG (C16:3)	[23]
**36**	511	533	8.96	\	Fr. 17	MGMG (C16:2)	[23]
**37**	537	559	9.86	\	Fr. 17	MGMG (C18:3)	[24]
**38**	539	561	11.24	\	Fr. 17	MGMG (C18:2)	
**39**	609	607	18.50	408	Fr. 17,18	15-Hydroxy-pheophorbide a *^j^*	[14]
**40**	929	951	19.90	\	Fr. 18	DGDG *^l^*(16:4/18:3)	[12]
**41**	931	953	20.89	\	Fr. 18	DGDG (16:3/18:3)	[12]
**42**	625	623	17.99	400	Fr. 19,20	Hydro-pheophorbide-lactone a	[25]

*^a^* Precursor ion in ESI (+); **1**, **2**, **3**, **4**, **11**, **24**, **26**, **27**, **28**, **33**, **39**, **42** exhibit [M + H]^+^ in ESI (+); **5**, **6**, **7**, **8**, **9**, **10**, **12**, **13**, **14**, **15**, **16**, **17**, **20**, **21**, **22**, **23**, **24** exhibit [M − H_2_O + H]^+^ in ESI (+); **31**, **32**, **34**, **35**, **36**, **37**, **40**, **41** exhibit [M + Na]^+^ in ESI (+). *^b^* Precursor ion in ESI (−); **1**, **2**, **3**, **4**, **31**, **32**, **34**, **35**, **36**, **37**, **40**, **41** exhibit [M + HCOO]^−^ in ESI (+); **5**, **6**, **7**, **8**, **9**, **10**, **11**, **12**, **13**, **14**, **15**, **16**, **17**, **20**, **21**, **22**, **26**, **27**, **28**, **33**, **39**, **42** exhibit [M − H]^−^ in ESI (+). *^c^* Retention time. *^d^* Column fraction(s) in which compounds are located. *^e^* Identification was confirmed by GC-MS analysis. *^f^* Fatty acid. *^g^* Fatty acid amide. *^h^* Compound forming a radical molecular ion [M]^+^. *^i^* Monogalactosyldiacylglycerol. *^j^*
**33** and **39** were identified as diastereoisomers. *^k^* Monogalactosylmonoacylglycerol. *^l^* Digalactosyldiacylglycerol.

**Table 2 molecules-25-01048-t002:** In vitro anti-inflammatory activities of compounds **19**, **21**, **26**, and **31**.

Compound	iNOS (IC_50_, µM)	NF-κB (IC_50_, µM)
19 *^a^*	26.4 ± 5.3	NA *^b^*
21	22.4 ± 2.4	122.3 ± 20.3
26	0.24 ± 0.03	132.1 ± 3.4
31	17.4 ± 1.3	47.0 ± 6.7
Parthenolide	0.72 ± 0.08	3.83 ± 0.60

*^a^* Obtained from a commercial source. *^b^* Not active at the highest test concentration of 87.9 μM (50 µg/mL).

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
