# Peer review of "Chemometrics-Assisted Identification of Anti-Inflammatory Compounds from the Green Alga Klebsormidium flaccidum var. zivo"

_molecules, 2020, doi:10.3390/molecules25051048_

Round 1
Reviewer 1 Report
review attached

Author Response
Reviewer 1
- The use of µM rather than µg/mL for IC50 values both in the abstract and throughout the manuscript may be more useful to readers for direct comparisons with other reported activities.
We have converted IC50 values from µg/mL into µM for all purified compounds in the abstract and throughout the manuscript. The changes have been highlighted in yellow in the revised manuscript.
- It is noted that the order in which compounds are discussed in the text is not the order in which they are numbered. How was the order in Table 1 selected? Why not list and number them in the order in which they are discussed in the text?
The compounds listed in Table 1 were numbered according to the order of fractions (Fr.1 to Fr.20) eluted from normal-phase column chromatography. For example, compound 1 is in Fr.1, and compound 42 in Fr. 19 and Fr. 20. This order provides relative polarity information for these compounds. In the text, the compounds are discussed following chemotypes of fatty acids, chlorophyll-derived compounds, carotenoids and glycerolipids. To address this reviewer comment, which is really helpful, we have included the compound numbers for all 42 compounds in the first paragraph of section 2.2 (lines 93-94) when the compounds are first mentioned. This has avoided the order issue of the compounds when they appear in the following text.
- p1, line 35, repeated use of ‘techniques’, “…chromatographic techniques instrumentation represented by hyphenated techniques…”
We have changed the text as suggested by the reviewer.
- p1, line 37, chemometrics is a scientific discipline described as the application of statistics to the field of chemical analysis. It is not a data analysis tool per se
We have changed the text from “data analysis tools such as chemometrics” to “data analysis tools utilized in metabolomics and chemometrics”.
- p1, line 39, ‘search’ should be ‘searching’: “…screening of active compounds [6], and searching for appropriate chemical markers for quality control [7].”
We have changed to the text as suggested by the reviewer.
- p7, line 176, and in Figure S6 legend in supplementary data: ‘maker’ should be ‘marker’.
We have corrected these errors as pointed out by the reviewer.
- p8, line 224: ‘BrukerAvance’ should be ‘Bruker Avance’
We have corrected this error as pointed out by the reviewer.
- p8, line 226: ‘Herts’ should be ‘Hertz’
We have corrected this error as pointed out by the reviewer.
- p9, line 246: “…prepare the water extract…”
We have changed to the text as suggested by the reviewer.
- p9, line 259: “…was were utilized for chemical profiling.”
We have corrected this error as pointed out by the reviewer.
- p9, line 260: “…Agilent Eclipse…”
We have corrected this typo.
- p9, line 272: “…acquired from in the ESI (+) mode…”
We have corrected this as suggested by the reviewer.
- p9, line 284: “…mouse macrophage cell line (RAW264.7) and human chondrosarcoma (SW1353) cell lines, respectively
We have changed the text as suggested by the reviewer (now in line 287).
- Table S1: What is the difference between ‘NA’ and ‘>50’? What is the thresh hold for NA? In figure 1 of the main text, the thresh hold is drawn at 30. Thus, surely anything 30-50 is ‘NA’.
In Table S1: The samples showing no inhibition of activity are marked as NA meaning no activity at 50 µg/mL. The samples that showed a % inhibition of 30% or more but did not reach a 50% inhibition are marked as >50 meaning that their IC50s could be reached if tested at >50 µg/mL. We have included this information in Table S1.
In Figure 1: we intentionally defined the threshold of IC50s at 30 just for the purpose of chemometrics analysis.
- Figure S1: add ‘ESI(-)’ to the lower panel.
We have made the correction as suggested by the reviewer.
Reviewer 2 Report
Review Molecules
The manuscript titled “Chemometrics-assisted identification of anti- inflammatory compounds from the green alga Klebsormidium flaccidum var. zivo’
The investigation describes a modified bio-guided fractionation of green alga to identified compounds with nitric oxide synthase activity based on uHPLC-qMS profiling using OPLS-DA analysis. They demonstrated that the main active compound pheophorbide a, an understudied natural product. It was found to inhibit iNOS and nuclear factor kappaB activation in phorbol 12-myristate 13-acetate-induced chondrocytes. The resultant findings indicate that this natural product warrants further evaluation.
Overall, this is a comprehensive isolation study that would fit in the readership of this journal. It provides the foundation for generating complex cyclic systems with promising biological properties that should be followed up. There are a few items that need to be addressed:
1) The following recent publications should be included as they also describe similar cascade reactions:
2) section 2.4 needs to be reworded. Lines 200-211 are particularly not clear.
3) does figure 2 have statistic analysis? Is such data reproducible for the same extract?
4) can you provide more information on the natural product pheophorbide a and include some recent work that might perhaps involved in the MoA of this compound (Aubry S, Fankhauser N, Ovinnikov S, et al. Pheophorbide a May Regulate Jasmonate Signaling during Dark-Induced Senescence. Plant Physiol. 2020;182(2):776–791. doi:10.1104/pp.19.01115)
5) Parthenolide is in the clinic but it is quite toxic, can you provide the relative toxicity of compound 26 with this control.
Author Response
Reviewer 2:
- The following recent publications should be included as they also describe similar cascade reactions:
This comment is not very clear to us as suggested publications were not provided. But we think we have included relevant, representative publications to support discussion.
- section 2.4 needs to be reworded. Lines 200-211 are particularly not clear.
We have carefully checked this section especially the statements in lines 200-211. We did not find any inappropriate statements from our perspective. In the revised manuscript, we inserted the wording Table 2 at the end of the sentence describing the activity of lutein (compound 19) in line 202. We hope this will help. We request the editor read and check it. If revision is needed, we will do so.
- does figure 2 have statistic analysis? Is such data reproducible for the same extract?
Yes, Figure 2 has showed the results of statistic analysis. We applied OPLS-DA to analyze active and inactive fractions using the SIMCA-P software. The score-plot of OPLS-DA analysis was shown in Figure 1. The S-plot and VIP-plot shown in Figure 2 were also derived from the OPLS-DA analysis. These results are totally reproducible for the same extract.
- can you provide more information on the natural product pheophorbide a and include some recent work that might perhaps involved in the MoA of this compound (Aubry S, Fankhauser N, Ovinnikov S, et al. Pheophorbide a May Regulate Jasmonate Signaling during Dark-Induced Senescence. Plant Physiol. 2020;182(2):776–791. doi:10.1104/pp.19.01115)
We appreciate the reviewer raise this comment and provide a reference showing pheophorbide a is a signaling molecule of chloroplast metabolic pathway in plants. As described in our manuscript, pheophorbide a demonstrated anti-inflammatory activity targeting iNOS and NF-kB in the present study, which had also been described in previous studies (references #27 and #31). We did a quick literature search and did not find critical references dealing with molecular targets of this compound.
- Parthenolide is in the clinic but it is quite toxic, can you provide the relative toxicity of compound 26 with this control.
We tested the cytotoxicity of compound 26 and parthenolide for the mouse macrophage cell line (RAW264.7). Compound 26 inhibited cell growth with an IC50 of 2.2 ug/mL, while parthenolide had an IC50 of 1.6 ug/mL. Based on these observations the toxicity of these two compounds seems to be very similar.
We also noticed that compound 26 (pheophorbide a) can be used as a photodynamic agent, and when rats were fed with pheophorbide a for photosensitization; the LD50 was 45.5 mg/100 g (Miki K et al, Nippon Nogei Kagaku Kaishi 1980, 54: 721-6). Pheophorbide a was reported to possess anti-tumor-promoting activitin mice (Nakamura Y et al. Cancer Letters 1996, 108: 247-255). These reports provide some information about its toxicity. We do appreciate the reviewer’s in-depth question, but this is not the focus of our manuscript.
Reviewer 3 Report
The manuscript by Shi Qiu et al is focused in Identification of Anti-inflammatory Compounds from Klebsormidium flaccidum algae. After close evaluation of manuscript I would suggest minor revision according to next points:
Fig.1 - at which wavelenght the chromatogram was recorded? In lines 200-201 - please clarify have you tested commercial lutein or isolated compound 1`9 (see Table 2) Section 3.8 - the samples preparation procedure should be described befora anti0inflammatory assays. In Table S2 authors have included 17 FA, while in Fig. S2 TIC include 18 FAMES. It wiull be helpful if authors indicate all FAME 37 standard (see Fig. S2).Author Response
Reviewer 3:
- Fig.1 - at which wavelenght the chromatogram was recorded?
The chromatogram shown in Figure 1 was generated from positive ESI-MS, which is indicated on the top of the chromatogram (right corner) as well as in the figure legend. For UV detection scan, the wavelength was recorded from 190 to 600 nm.
- In lines 200-201 - please clarify have you tested commercial lutein or isolated compound 1`9 (see Table 2)
We used the commercial lutein for bioassay as indicated in lines 200-201. To make it clear, we inserted the wording (Table 2) in line 202, and added a footnote under Table 2 indicating compound 19 is a commercial sample.
- Section 3.8 - the samples preparation procedure should be described before anti-inflammatory assays.
We have included a brief sample preparation procedure at the beginning of this section (lines 284-285).
- In Table S2 authors have included 17 FA, while in Fig. S2 TIC include 18 FAMES. It will be helpful if authors indicate all FAME 37 standard (see Fig. S2).
We have included the identification of FAME 37 standard in Figure S2. We have corrected our labeling for the 17 fatty acids in the revised Figure S2, as one compound (labeled as compound 12 in the original Figure S2) is phytol (retention time 37.844 min), which is not labeled in the revised Figure S2. There are no changes for Table S2 as the compound identification is correct.